# Genetic alterations in the 3q26.31-32 locus confer an aggressive prostate cancer phenotype

Benjamin S. Simpson [1], Niedzica Camacho[2,3,4], Hayley J. Luxton [1], Hayley Pye [1], Ron Finn[1], Susan Heavey [1], Jason Pitt[5], Caroline M. Moore[6] & Hayley C. Whitaker [1✉]

Large-scale genetic aberrations that underpin prostate cancer development and progression, such as copy-number alterations (CNAs), have been described but the consequences of specific changes in many identified loci is limited. Germline SNPs in the 3q26.31 locus are associated with aggressive prostate cancer, and is the location of *NAALADL2*, a gene over-expressed in aggressive disease. The closest gene to *NAALADL2* is *TBL1XR1*, which is implicated in tumour development and progression. Using publicly-available cancer genomic data we report that *NAALADL2* and *TBL1XR1* gains/amplifications are more prevalent in aggressive sub-types of prostate cancer when compared to primary cohorts. In primary disease, gains/amplifications occurred in 15.99% (95% CI: 13.02–18.95) and 14.96% (95% CI: 12.08–17.84%) for *NAALADL2* and *TBL1XR1* respectively, increasing in frequency in higher Gleason grade and stage tumours. Gains/amplifications result in transcriptional changes and the development of a pro-proliferative and aggressive phenotype. These results support a pivotal role for copy-number gains in this genetic region.

[1] Molecular Diagnostics and Therapeutics Group, Research Department of Targeted Intervention, Division of Surgery & Interventional Science, University College London, London, UK. [2] Human Oncology and Pathogenesis Program, Memorial Sloan Kettering Cancer Center, New York, NY, USA. [3] Marie-Josée and Henry R. Kravis Center for Molecular Oncology, Memorial Sloan Kettering Cancer Center, New York, NY, USA. [4] Department of Pathology, Memorial Sloan Kettering Cancer Center, New York, New York for Genomics Research, Discovery Sciences, Biopharmaceutical R&D, AstraZeneca, Cambridge, UK. [5] Cancer Institute of Singapore, National University of Singapore, Singapore, Singapore. [6] Department of Urology, UCLH NHS Foundation Trust, London, UK. ✉email: Hayley.Whitaker@ucl.ac.uk

Prostate cancer (PCa) is the most common non-cutaneous cancer in developed countries[1,2] and is defined by dynamic genome alterations and both its pathological and genetic heterogeneity[3]. An important pathological predictor of prostate cancer aggressiveness is Gleason grade, used to assess risk of progression and stratify patients for treatment, however, the underlying genomic changes which accompany more aggressive tumours remains incompletely defined.

Overall copy-number alteration (CNA) burden has been linked to poorer prognosis in prostate cancer, associating with Gleason grade, biochemical recurrence and prostate cancer specific death, however the exact mechanism driving these prognostic changes is unknown and thought to be primarily driven by general chromosomal instability[4–6]. Changes in specific loci have also been linked to aggressiveness, in particular gains in proliferative genes e.g. MYC (8q24) and loss of tumour suppressors PTEN (10q23) and NKX3-1 (8p21)[7,8]. Many genetic alterations have been linked with prostate cancer such as point mutations in SPOP, FOXA1, and IDH1[9]. Large-scale oncogenic structural rearrangements, translocations and copy-number changes are also common, often leading to the coordinated dysregulation of multiple elements for example the loss of 21q, which is associated with the TMPRSS: ERG fusion rearrangement and the subsequent rearrangement of SMAD4[10]. Improved understanding of the mechanisms governing disease pathogenesis and progression may allow for better therapeutic exploitation, for example genetic alterations in the DNA repair machinery have been linked to susceptibility to PARP inhibitors in a range of tumour types and alterations in AR confer sensitivity or resistance to androgen deprivation therapy in metastatic castrate resistant prostate cancer (mCRPC)[11].

NAALADL2 is located on 3q26.31 and is a member of the glutamate carboxypeptidase II family along with the widely studied PCa marker PSMA (NAALAD1)[12], and its expression has previously been associated with prostate tumour stage and grade[13] with expression predicting poor survival following radical prostatectomy[13]. A large genome-wide association study (GWAS) of 12,518 prostate cancer cases found rs78943174, a SNP within the 3q26.31 (NAALADL2) locus was associated with high Gleason sum score[14]. A further rs10936845 SNP was identified within a GATA2 motif that increases NAALADL2 expression in prostate cancer patients, where increased expression also predicted biochemical reccurence[15]. The same study showed even higher binding preference to HOXB13 and FOXA1 to this site, suggesting

co-occupancy by these important transcription factors, both of which have been shown to be involved in AR cistrome reprogramming[15,16].

Adjacent to NAALADL2 in the genome is TBL1XR1, a core component of nuclear receptor corepressor (NCoR) complex, that acts as a coregulator of nuclear receptors, influencing several cellular functions, including: growth, anti-apoptosis, and inflammation[17]. TBL1XR1 is also an androgen receptor (AR) co-activator[18]. Expression of TBL1XR1 has been associated with poor prognosis in several cancers, predicting poor overall survival and lymph node metastasis in gastric[19] and ovarian cancers[20] and recurrence in colorectal[21], breast[22] and liver cancers[23].

Here we utilise large-scale publicly available genomic data to better characterise the broad somatic copy-number changes occurring within the 3q26.31-32 locus, particularly centred around gains/amplifications in NAALADL2 and TBL1XR1 and linking them to the clinical characteristics of aggressive prostate cancer.

## Results

**3q26.31-32 gain frequency is increased in aggressive PCa.** Copy-number alterations often alter the expression of the gene in which they occur with gene dosage known to correlate with mRNA expression. Genetic structural variants are also known to alter transcriptional regulation by altering cis-regulatory elements such as promotors and enhancers, resulting in differential expression[24,25]. Increased NAALADL2 and TBL1XR1 expression have previously been linked to poor prognosis in cancers leading us to examine the frequency of somatic copy-number gains in these genes across various prostate cancer subtypes[19–21,26]. Alteration frequency was assessed using data from cBioportal (Fig. 1a) and all study data was processed using a standardised pipeline to ensure comparable results. Alteration frequency was assessed in a total of 3804 patients (4029 samples) in 16 non-overlapping studies (Appendix 1); eleven studies focused on primary prostate cancer, four on metastatic prostate cancer and one on neuroendocrine and castrate-resistant cancers. Significant copy-number increases above a derived background threshold were categorised as 'gains' and copy-number decreases as 'deletions'. Overall, the distribution of NAALADL2 and TBL1XR1 alterations were significantly different between disease sub-types to that which is expected (Chi-squared goodness-of-fit test:

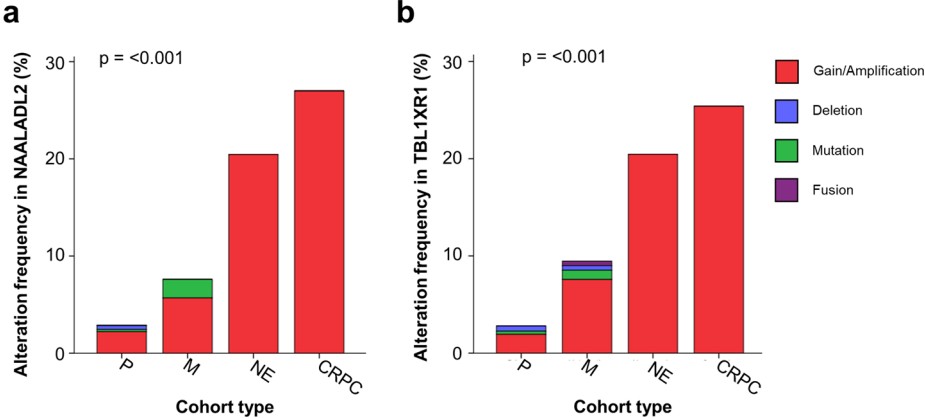

**Fig. 1 Somatic alteration frequency of NAALADL2 and TBL1XR1 across prostate cancer subtypes in publically available genomic studies (n = 3804).** **a** NAALADL2 genetic alteration frequency (%) across different subtypes of prostate cancer. **b** TBL1XR1 genetic alteration frequency (%) across different subtypes of prostate cancer. P = primary prostate cancer, M = metastatic prostate cancer, NE = neuroendocrine prostate cancer and castrate resistant prostate cancer (CRPC). All annotations were assigned using Genome Nexus and CNAs are called using GISTIC or RAE algorithms. P-values show the results of a Chi-squared goodness-of-fit test to determine if the number of observed patients with each alteration type is different from that which is expected across each cancer subtype. Results detailed in Supplementary data 1.

$p = 1.19 \times 10^{-6}$ and $p = 2.39 \times 10^{-6}$, Fig. 1a, b), with gains being most frequent in castrate-resistant prostate cancer (26.98% and 25.4% respectively), followed by neuroendocrine (20.45% and 20.45%), metastatic (5.69% and 7.58%) then primary prostate cancer (2.22% and 1.93%, Fig. 1a, b).

**3q26.31-32 gains extend across an oncogene-rich region of Chr3.** As CNA's are known to associate with more aggressive subtypes of prostate cancer, we investigated their association with clinical characteristics to establish if changes can be detected early in the life history of cancer, predicting more aggressive disease. We utilised copy-number data from primary organ confined disease from both the UK and Canadian International Cancer Genome Consortium (ICGC) cohorts and The Cancer Genome Atlas (TCGA). These studies use intermediate-high risk prostate cancer patients with no treatment prior to radical prostatectomy. To allow comparisons between the studies, data were re-analysed using the Genomic Identification of Significant Targets in Cancer 2 (GISTIC2) method, favoured by the broad institute and TCGA[27] as it distinguishes between low-level copy number increases (gains) and high-level copy-number increases (amplifications). Within the three cohorts, we found that copy-number gains across both genes were frequent, with gains in *NAALADL2* ranging from 6.53% (Canada) to 15.63% (UK) and between 0–2.4% for amplifications (Table 1). *TBL1XR1* had an almost identical CNA frequency of between 6.01% (UK) to 15.63 (Canada, Table 1).

We fitted a random-effects model to more accurately estimate the frequency of *NAALADL2* and *TBL1XR1* gain/amplifications combining the data from all three cohorts, which estimated the true frequencies to be 13.06% (95% CI: 7.85–18.27%) and 12.29% (95% CI: 7.11–17.47%) for *NAALADL2* and *TBL1XR1* respectively (Supplementary Fig. 1). Leave-one out analysis and a diagnostic plots revealed that the ICGC Canada study was a significant source of heterogeneity therefore, the study was removed and the model refit. The final estimated frequency of *NAALADL2* and *TBL1XR1* gains/amplifications was 15.99% (95% CI: 13.02–18.95) and 14.96% (95% CI: 12.08–17.84%) respectively in primary prostate cancer.

Due to their close proximity in the genome we investigated if gains/amplifications in *NAALADL2* and *TBL1XR1* co-occurred in the same patients using a genome-wide Fisher's exact test with a false discovery rate correction. *NAALADL2* and *TBL1XR1* significantly co-amplified in all three cohorts; ICGC UK $p = 2.31 \times 10^{-12}$, ICGC Canada $p = 1.58e \times 10^{-34}$ and TCGA $p = 1.90 \times 10^{-10}$, (Supplementary Fig. 2). Additionally, testing confirmed that wide-spanning gains/amplifications occurred in neighbouring regions in the majority of patients. In the ICGC UK

cohort ($n = 99$), there was a significant co-occurrence of somatic copy-number gains/amplifications in *NAALADL2* with *TBL1XR1* (FDR-corrected Fisher's exact test = $1.01 \times 10^{-09}$, Fig. 2a). Gains in this region also significantly correlated with two regions spanning chromosomes 7 and 8, both gains previously described as being abundant in prostate cancer (Supplementary Data 2)[28]. The Canadian cohort ($n = 389$) showed a similar pattern of co-occurrence with gains/amplifications spanning the region surrounding *NAALADL2* and *TBL1XR1* (3p25.3 to ~3q29, Fig. 2b). There was also a significant co-occurrence with gains in the beginning of chromosome 4 as well as some sporadic co-occurrence across the genome (Fig. 2b, Supplementary Data 2). These results were supported by the outcome of the same analysis in the TCGA cohort ($n = 498$), although several large spikes of co-occurrence were also observed in regions not local to *NAALADL2* and *TBL1XR1*, as these spikes were not present in the other two cohorts they most likely represent artefacts (Fig. 2c, Supplementary Data 2). Overall, across the three cohorts, there were was a consistent co-amplification in region spanning 465 genes between 3p14.1 and 3q29. While a number of patients had multiple CNAs, we found no consistent co-occurrence with common CNAs such as *MYC* gain, *FGFR1* gain, *PTEN* loss, *RB1* loss or *NKX3-1* loss (FDR-corrected Fisher's exact test: $p > 0.05$).

The 3q26 region where *NAALADL2* and *TBL1XR1* are located is rich in oncogenes such as *PIK3CA*, *SOX2*, *ECT2* and *PRKCI* which may act to drive tumorigenesis[29]. We determined the number of known oncogenes within this defined region by comparing the 465 overlapping genes that co-amplified with *NAALADL2* and *TBL1XR1* in all three cohorts, against the Network of Cancer Genes database[30]. This revealed that 67 (14.09%) of genes are known oncogenes including *BCL6*, *ATR* and *PI3K* family members (Supplementary Data 3). These results confirm that a high proportion of prostate cancer patients develop large copy-number gains across multiple oncogenes in this genetic region.

**Gains in 3q26.31-32 associate with adverse clinical features.** Common prostate cancer CNAs, such as those in *MYC* and *PTEN*, are known to associate with higher Gleason grade[31]. Consistent with these findings, we also found *NAALADL2* and *TBL1XR1* amplifications were highly correlated with Grade Group (GG), showing that the frequency of *NAALADL2* and *TBL1XR1* gains tripling between GG1 and GG2 lesions and more than doubling between GG2 and 3 (Table 2). A Chi-squared goodness-of-fit test showed that the distribution of gains/amplifications between Grade groups was significantly different to the distribution of diploid patients for both *NAALADL2* and *TBL1XR1* ($p = 7.844 \times 10^{-08}$ and $p = 9.179 \times 10^{-08}$). When

**Table 1 Alteration frequency of *NAALADL2* and *TBL1XR1* called via the GISTIC2 method in three non-overlapping primary, organ-confined, radical prostatectomy cohorts from the International Cancer Genome Consortium (ICGC) and The Cancer Genome Atlas (TCGA).**

| | ICGC UK | | | | ICGC CANADA | | | | TCGA | | | |
|---|---|---|---|---|---|---|---|---|---|---|---|---|
| | NAALADL2 | | TBL1XR1 | | NAALADL2 | | TBL1XR1 | | NAALADL2 | | TBL1XR1 | |
| Alteration | n | % | n | % | n | % | n | % | n | % | n | % |
| Deep Del | 0 | 0.00% | 1 | 1.04% | 0 | 0.00% | 0 | 0.00% | 6 | 1.22% | 7 | 1.42% |
| Shallow Del | 4 | 4.17% | 5 | 5.21% | 5 | 1.31% | 8 | 2.09% | 9 | 1.83% | 14 | 2.85% |
| Diploid | 77 | 80.21% | 75 | 78.13% | 345 | 90.08% | 345 | 90.08% | 398 | 80.89% | 398 | 80.89% |
| Gain | 15 | 15.63% | 15 | 15.63% | 25 | 6.53% | 23 | 6.01% | 64 | 13.01% | 60 | 12.20% |
| Amplification | 0 | 0.00% | 0 | 0.00% | 8 | 2.09% | 7 | 1.83% | 15 | 3.05% | 13 | 2.64% |
| Total | 96 | 100.00% | 96 | 100.00% | 383 | 100.00% | 383 | 100.00% | 492 | 100.00% | 492 | 100.00% |

The degree of copy number alteration is discretised into five categories: amplification, gain (representing low and high level copy number increase), diploid (no significant CNA) and shallow and deep deletion (representing low and high level copy number loss).

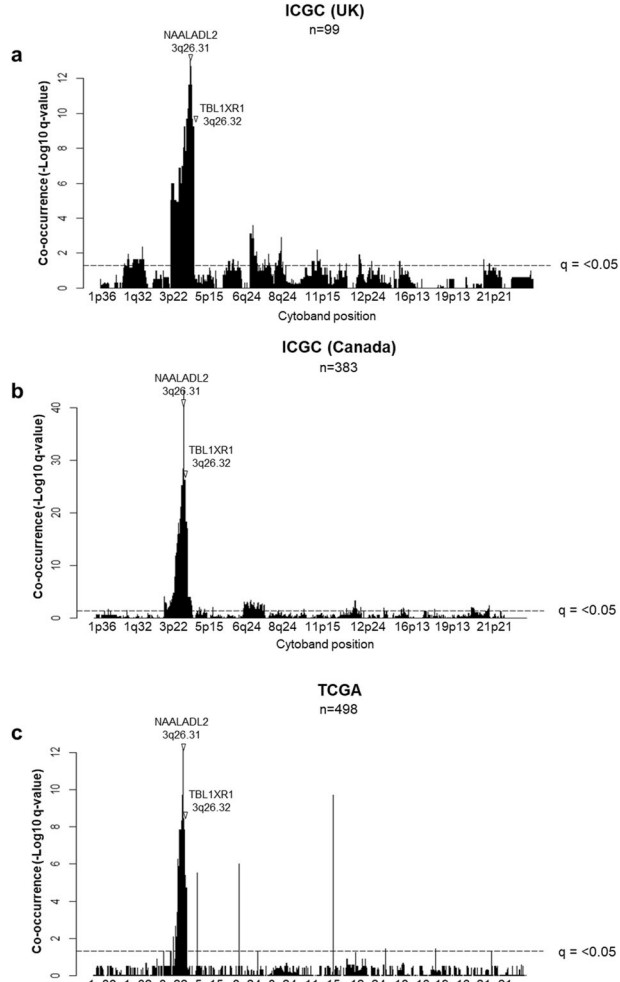

**Fig. 2 Genome-wide co-occurrence with NAALADL2 and TBL1XR1 gains/ amplifications.** The Y axis shows −log10 q-values from a Fishers exact test between gain/amplifications in NAALADL2 and co-occuring genes. The dotted line represents the threshold for statistical significance after correction for multiple testing. **a** Significantly co-occurring gains across the genome in the ICGC UK cohort. **b** Significantly co-occurring gains across the genome in the ICGC Canada cohort. **c** Significantly co-occurring gains across the genome in the TCGA cohort. NAALADL2 and TBL1XR1 cytoband positions are labelled. All Fisher tests use NAALADL2 gain or amplification as the altered group. Full results are detailed in Supplementary Data 2.

compared to common CNAs such as PTEN loss and MYC gain, the alteration frequency of NAALADL2 and TBL1XR1 was more correlated with higher Gleason grade groups; Spearmans rho was 0.9 ($p = 0.035$), 0.9 ($p = 0.035$) for NAALADL2 and TBL1XR1 and 0.6 ($p = 0.28$), 0.7 (0.19) for PTEN and MYC respectively (Supplementary Fig. 3A).

Moreover, we also noted the same pattern of increasing frequency of gains with T stage (Chi-squared goodness-of-fit test: $p = 0.00041$ and $p = 0.0028$ respectively, Table 3).

Patients with gains exhibited differences in the location of the tumour within the prostate, with 57.69% and 57.14% of those with NAALADL2 and TBL1XR1 gains having tumours in overlapping and multiple zones compared to just 42.92% and 43.24% for those without gains (Chi-squared goodness-of-fit test: $p = 0.015$ and $p = 0.012$. There was also an increased relative number of positive surgical margins (Chi-squared goodness-of-fit test: $p = 0.0013$ and $p = 0.00059$ in patients with gains (48.00%

and 47.82%) compared to those without (26.93% and 27.27%). Moreover, of the 401 patients who had their lymph nodes examined, the percentage of patients with lymph node positivity defined through positivity on haematoxylin and eosin staining (H&E) was more than double in patients with NAALADL2 or TBL1XR1 gains (32.35% and 34.92%) compared to those without a gain (16.82% and 16.57%, Chi-squared goodness-of-fit test: $p = 0.0032$ and $p = 0.00073$. Finally, while only one man in the cohort had evidence of positive findings in his bone scan, we did observe a significant between the number of equivocal bone scans in patients with gains: 8.16% and 8.69% compared to 2.09% and 2.06% in those patients without gains (Chi-squared goodness-of-fit test: $p = 0.011$ and $p = 0.0058$ for NAALADL2 and TBL1XR1 respectively), however, the number of expected cases in each of these categories was less than 5, adding some uncertainty to this result. We found no significant difference in the mean age between patients with different copy-numbers of NAALADL2 or TBL1XR1 (Kruskall-Wallis rank sum test: $p = 0.12$ and 0.23).

As gains/amplifications in NAALADL2 and TBL1XR1 coincide with a cluster of known oncogenes and coincide with clinical variables linked to more aggressive disease, we also compared disease-free survival. Comparing patients with gains/amplifications in NAALADL2 and TBLXR1 to those with diploid copies, we observed no significant association in the ICGC UK cohort ($n = 99$), although there was a trend towards reduced disease-free survival (Supplementary Fig. 4A). In the larger TCGA cohort ($n = 498$) there was a significant reduction in disease-free survival in patients with a gain in either NAALADL2 (Log-rank Mantel-Cox: $p = 0.019$) or TBL1XR1 (Log-rank Mantel-Cox: $p = 0.024$, Supplementary Fig. 4B).

Univariable Cox regression confirmed that carrying a gain/ amplification in NAALADL2 and TBL1XR1 in the TCGA cohort resulted in reduction in disease-free survival, hazard ratio (HR): 1.73 (95% CI: 1.091–2.914, $p = 0.021$). For reference, we performed a similar analysis of patients with PTEN deletion or MYC gains, two common copy number alterations with proven association with disease-free survival in prostate cancer[32,33]. When patients were stratified solely by CNA status and survival compared using the Kaplan-Meier method, those patients with MYC gain or PTEN deletion (homo or hemizygous) showed no significant difference in disease-free survival (Log-rank Mantel-Cox: $p = 0.11$ and $p = 0.077$ respectively) while those stratified by NAALADL2 gain, TBL1XR1 gain or both NAALADL2 and TBL1XR1 gain showed significant differences in survival (Log-rank Mantel-Cox: $p > 0.024$), Supplementary Fig. 5A–E). Univariable Cox regression estimated the hazard ratios for these copy-number alterations as: 1.415 (95% CI: 0.9256–2.163), 1.458 (95% CI: 0.9575–2.22), and 1.897 (95% CI: 1.15–3.131) for MYC, PTEN and NAALADL2/TBL1XR1 respectively. We also compared the disease-free survival of patients with only a copy-number alteration in each of the four genes where each group was mutually exclusive (Supplementary Fig. 5F, G). This showed that on the whole, patients with CNAs in NAALADL2/TBL1XR1 had reduced or equal disease-free survival as those with either only MYC gain or only PTEN loss. Patients with copy number gains in both had a worse prognosis. All clinical data is available in Supplementary Data 4.

As CNAs in NAALADL2 and TBL1XR1 were associated with clinical characteristics such as Gleason grade group and T stage, we used multivariable Cox regression models to confirm that any changes in survival were driven by these associations and found that copy number gains in NAALADL2 and TBL1XR1 were no longer significant once corrected for Gleason grade and T stage ($p = 0.71184$, Supplementary Data 5). These results suggest that the differences in disease-free survival seen when stratified by

**Table 2 The frequency of NAALADL2 and TBL1XR1 gain/amplifications by Gleason Grade Group in the TCGA cohort.**

| | NAALADL2 | | TBL1XR1 | | |
|---|---|---|---|---|---|
| Grade group | Diploid | Gain | Diploid | Gain | Total |
| GG1 | | | | | |
| Observed | 43 | 2 | 44 | 1 | 45 |
| Expected | 37.8 | 7.2 | 38.3 | 6.7 | 45 |
| % within GG | **95.60%** | **4.40%** | **97.80%** | **2.20%** | **100.00%** |
| GG2 | | | | | |
| Observed | 136 | 9 | 136 | 9 | 145 |
| Expected | 121.7 | 23.3 | 123.4 | 21.6 | 145 |
| % within GG | **93.80%** | **6.20%** | **93.80%** | **6.20%** | **100.00%** |
| GG3 | | | | | |
| Observed | 86 | 13 | 87 | 12 | 99 |
| Expected | 83.1 | 15.9 | 84.3 | 14.7 | 99 |
| % within GG | **86.90%** | **13.10%** | **87.90%** | **12.10%** | **100.00%** |
| GG4 | | | | | |
| Observed | 51 | 12 | 53 | 10 | 63 |
| Expected | 52.9 | 10.1 | 53.6 | 9.4 | 63 |
| % within GG | **81.00%** | **19.00%** | **84.10%** | **15.90%** | **100.00%** |
| GG5 | | | | | |
| Observed | 96 | 43 | 98 | 41 | 139 |
| Expected | 116.6 | 22.4 | 118.3 | 20.7 | 139 |
| % within GG | **69.10%** | **30.90%** | **70.50%** | **29.50%** | **100.00%** |
| **Total (n)** | 412 | 79 | 418 | 73 | 491 |

Grade groups defined as: Grade Group 1 = Gleason score ≤6, Grade Group 2 = Gleason score 3 + 4 = 7, Grade Group 3 = Gleason score 4 + 3 = 7, Grade Group 4 = Gleason score 8, Grade Group 5 = Gleason scores 9 and 10. Displayed are the numbers of patients (observed) with (gain) or without (diploid) a gain/amplification in this region in each Grade Group. Additionally the expected number of patients estimated to be within each category is also shown, along with the percentage of each Grade Group which is made up by patients with or without a gain. Bold values indicate the overall percentage of the group with a given copy-number state. All clinical data detailed in Supplementary Data 4.

**Table 3 The frequency of NAALADL2 and TBL1XR1 gain/amplifications by T stage in the TCGA cohort.**

| | NAALADL2 | | TBL1XR1 | | |
|---|---|---|---|---|---|
| T stage | Diploid | Gain | Diploid | Gain | Total |
| T2a | | | | | |
| Observed | 12 | 1 | 12 | 1 | 13 |
| Expected | 10.9 | 2.1 | 11.1 | 1.9 | 13 |
| % within T stage | **92.30%** | **7.70%** | **92.30%** | **7.70%** | **100.00%** |
| T2b | | | | | |
| Observed | 10 | 0 | 10 | 0 | 10 |
| Expected | 8.4 | 1.6 | 8.5 | 1.5 | 10 |
| % within T stage | **100.00%** | **0.00%** | **100.00%** | **0.00%** | **100.00%** |
| T2c | | | | | |
| Observed | 149 | 14 | 150 | 13 | 163 |
| Expected | 136.7 | 26.3 | 138.8 | 24.2 | 163 |
| % within T stage | **91.40%** | **8.60%** | **92.00%** | **8.00%** | **100.00%** |
| T3a | | | | | |
| Observed | 130 | 26 | 132 | 24 | 156 |
| Expected | 130.9 | 25.1 | 132.8 | 23.2 | 156 |
| % within T stage | **83.30%** | **16.70%** | **84.60%** | **15.40%** | **100.00%** |
| T3b | | | | | |
| Observed | 100 | 32 | 102 | 30 | 132 |
| Expected | 110.7 | 21.3 | 112.4 | 19.6 | 132 |
| % within T stage | **75.80%** | **24.20%** | **77.30%** | **22.70%** | **100.00%** |
| T4 | | | | | |
| Observed | 5 | 5 | 6 | 4 | 10 |
| Expected | 8.4 | 1.6 | 8.5 | 1.5 | 10 |
| % within T stage | **50.00%** | **50.00%** | **60.00%** | **40.00%** | **100.00%** |
| **Total (n)** | 406 | 78 | 412 | 72 | 484 |

Displayed are the numbers of patients (observed) with (gain) or without (diploid) a gain/amplification in this region in each T stage. Additionally the expected number of patients estimated to be within each category is also shown, along with the percentage of each T stage which is made up by patients with or without a gain. Bold values indicate the overall percentage of the group with a given copy-number state. All clinical data detailed in Supplementary Data 4.

gain/amplification status are driven by strong association with these clinical variables.

In the ICGC cohorts, individuals with somatic single-base alterations in *NAALADL2* also associated with reduced disease-free survival in a combined ICGC cohort as well as associating with reduced disease-free and overall survival in an early onset prostate cancer cohort (ICGC EOPC, Denmark). Single-base substitutions in *TBL1XR1* were only associated with disease-free survival in the ICGC EOPC cohort (Supplementary Fig. 6). Single base alterations did not occur with a frequency greater than one in any single base in *NAALADL2* or *TBL1XR1*.

**3q26.31-32 gains co-occur with pro-proliferative transcription**. To determine the potential functional consequences of gains within the *NAALADL2* and *TBL1XR1* amplicon, mRNA expression profiles were explored using the TCGA RNAseq data. DESeq2 was used to determine differentially expressed genes between patients with copy-number gains for both *NAALADL2* and *TBL1XR1*, compared to those without. For *NAALADL2* there were 4123 differentially expressed genes (DEGs) and 4091 DEGs for *TBL1XR1* when the two groups were compared (FDR < 0.05, Supplementary Data 6). Our previous study on *NAALADL2* identified nine genes which were reciprocally regulated by overexpression or knockdown of *NAALADL2*[26]. Of these nine we found that three (cancer antigen *XAGE1B*, adhesion/motiliy regulator *SPON2* and AR regulator *HN1*) were significantly differentially expressed (*p* < 0.022) in patients with a *NAALADL2* gain and in the same direction as the overexpression model[26,34–36].

When comparing the DEGs between patients with and without gains/amplifications in either *NAALADL2* or *TBL1XR1* we observed that 77.9% of the DEGs overlapped between *NAALADL2* and *TBL1XR1* (Fig. 3a). 48.8% (227/465) of the genes were located within the locus we identified as co-amplified with *NAALADL2* and *TBL1XR1* and were differentially expressed, consistent with a mechanism of self-regulating expression[24,25]. *TBL1XR1* was one of the significant overlapping DEGs, *NAALADL2* was just on the boundary of statistical significance (FDR corrected Wald test: *p* = 0.053, Supplementary Fig. 7).

*NAALADL2* has been shown to be co-expressed with number of androgen regulated proteins and contains a number of AR binding sites and *TBL1XR1* is an AR coactivator and may be involved in AR cistrome reprogramming[18,26,37,38]. We therefore looked at overlap between androgen regulated genes with AR binding sites (full or partial) and genes demonstrated to be androgen regulated following R1881 stimulation in at least two independent studies[37,39]. 50 shared genes were differentially expressed in patients with *NAALADL2* and *TBL1XR1* gains/amplifications that contained AR binding sites and demonstrated androgen regulation by R1881, 506 (14.09%) genes had either a AR binding motif, were androgen regulated in two or more studies, or both (Fig. 3b).

Of the overlapping DEGs, a total of 473 (13.15%) were known oncogenes (Supplementary Data 7) which may drive an aggressive clinical phenotype. Of note was PI3K family members: *PIK3C2G, PIK3CA, PIK3CB, PIK3R4*, Mucin family members: *MUC1, MUC4* and *MUC6* and other prostate cancer associated genes such as *SMAD4, SOX9* and *SPOP*[7,9,40,41]. Additionally several genes which form commercial prognostic assays were also differentially expressed, such as the Decipher assay (*NFIB, LASP1, ZWILCH, THBS2, COL1A2* and *COL5A1*)[42], Oncotype DX assay (*SFRP4, COL1A1, KLK2, TPX2*)[43,44] and the Prolaris assay (*ASPM, BUB1B, CENPF* and *FOXM1*)[45].

We inspected of the top 50 most significant shared DEGs using unsupervised hierarchal clustering (Fig. 3b, Supplementary Data 8). DEGs mostly displayed upregulation, consistent with a gene-dosage effect (Fig. 3b)[24]. Enrichment for biological processes was assessed by Gene-set enrichment analysis (GSEA) for *NAALADL2* and *TBL1XR1* gains separately, and by over-representation analysis (ORA) on the shared DEG list using WebGestalt[46].

GSEA on the individual lists of DEGs showed that, despite a large overlap, the enriched biological processes did differ between the two genes; patients with a gain in *NAALADL2* showed enrichment in processes related to NADH dehydrogenase complex assembly (FDR = 0.0030), mitochondrial respiratory chain complex assembly (FDR = 0.0060), translational initiation (FDR = 0.023), cytochrome complex assembly (FDR = 0.029343), protein localisation to endoplasmic reticulum (FDR = 0.035) and cytoplasmic translation (FDR = 0.037, Supplementary Data 9). Patients with a gain in *TBL1XR1* showed enrichment in mitotic cell cycle phase transition, chromosome segregation, actin filament-based movement, microtubule cytoskeleton organisation involved in mitosis, regulation of cell cycle phase transition, cell cycle G1/S phase transition (FDR < 0.0001), as well as a number of other processes (Supplementary Data 9).

To understand the combined effect of gains/amplification in these genes, we investigated overrepresentation of processes in the DEGs which were common to both *NAALADL2* and *TBL1XR1*. In the shared DEG list, the significantly enriched Gene Ontology (GO) biological processes were all involved in the cell cycle cycle pathway including: mitotic regulation and chromosome segregation (Fig. 3c, Supplementary Data 10). These findings support a hypothesis whereby gains in *NAALADL2* and *TBL1XR1* concomitantly bring about mRNA expression changes which support an aggressive pro-proliferative phenotype in primary prostate cancer.

## Discussion

In this study we present evidence that somatic copy-number gains in *NAALADL2* and *TBL1XR1* are more frequent in high grade and aggressive forms of prostate cancer. These results are bolstered by studies which have identified CNAs in this region in mCRPC, however, to our knowledge this is the first time these gains have been reported in neuroendocrine disease[47]. We also demonstrate that *NAALADL2* and *TBL1XR1* gains occur in an earlier setting, co-occurring with gains in neighbouring genes. A major barrier to the adoption of CNA based tests in the clinic is the reliance on expensive NGS approaches as well as sufficient sequencing depth and coverage to assess overall copy-number burden. The discovery of smaller clinically significant loci could allow for cheaper, quicker targeted approaches, particularly if a single loci can elude to gains/amplifications in a larger region.

In primary prostate cancer, Gains/amplifications in this region associated with Gleason grade, tumour stage, number of positive lymph nodes, bone scan results and as these variables contribute to time to disease-free survival, patients stratified by *NAALADL2/TBL1XR1* status also have altered disease-free survival times. Our work is supported by previous studies that have eluded to the clinical significance of this locus, particularly as germline SNPs within this locus have been associated with higher Gleason grade tumours and more aggressive disease[14]. This also supports smaller studies such as those by Heselmeyer-Haddad et al., who identified two out of seven patients with gains in *TBL1XR1* in recurrent prostate cancer[48]. However, these studies investigated these genes in isolation, naïve to the larger context in which these alterations occur. Here we have found that gains/amplifications at this locus not only co-amplify with other described oncogenes but associate with much larger transcriptional changes which are consistent with the observed aggressive clinical phenotype.

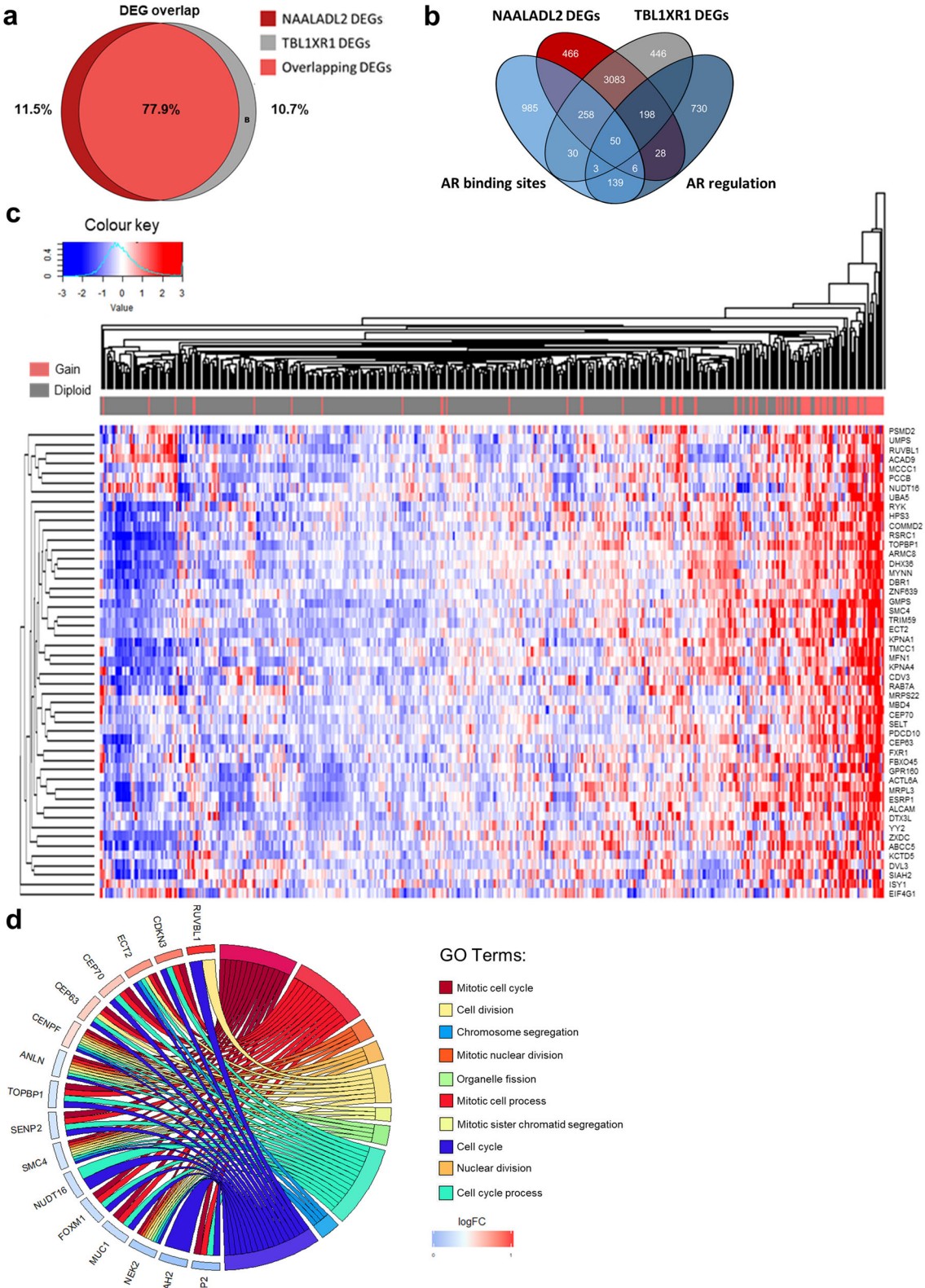

**Fig. 3 Transcriptomic changes in patients with NAALADL2/TBL1XR1 gains. a** Venn diagram showing the number and percentage of overlapping DEGs between patients with NAALADL2 gain/amplification and TBL1XR1 gain/amplification (77.9% overlap). **b** Venn diagram showing the number of NAALADL2 and TBL1XR1 DEGs and genes with identified AR binding sites (determined through ChIP-Seq and AR knockdown) and genes shown to be androgen regulated following R1881 stimulation. **c** Unsupervised hierarchal clustering of the top 50 most significant DEGs, bar beneath upper dendrogram shows copy-number status of patients where red is patients with a gain in both NAALADL2 and TBL1XR1 and grey represents those without gain/ amplification in these genes. Heatmap represents mean-centred z scores derived from RKPM values. **d** Chord diagram showing significantly over-represented GO biological processes and key genes within these processes. All clinical data detailed in Supplementary Data.

Overall changes in copy-number burden have been shown to be indicative of genetically unstable tumours and predict prostate cancer relapse[5]. Many single CNAs have already been described that predict PSA recurrence after radical prostatectomy including: *PTEN* loss, co-occurrence of *PTEN*, *FAS* (10q23.31) and *PAPSS2* (10q23.2–10q23.31) loss, a loss of 16q with or without a loss of *PTEN*, a loss within 6q, 13q, gains in MYC, 11q13.17, 7q, and a concurrent loss of 8p22 with a gain of 8q248[7–9,28,49]. Compared to well-known CNAs such as *PTEN* loss and *MYC* amplification, we have observed that Gains/amplifications in *NAALADL2/TBL1XR1* equally or better segregate patients who will have reduced disease-free survival.

The gains/amplifications in *NAALADL2/TBL1XR1* also corresponded to a significant increase in both *NAALADL2* and *TBL1XR1* mRNA, supporting previous studies that have described upregulation of these genes and linked them to poor prognosis in various cancers[19–21,26]. This suggests that gains in these genes may cause increased expression of *NAALADL2* and *TBL1XR1* in cancers. We also noted a number of the differentially expressed genes between patients with and without a gain/ amplification in *NAALADL2/TBL1XR1* have been shown to be androgen regulated, however further work is required to determine if gains/amplifications in this region cause changes in AR transcriptional regulation through cis regulatory elements or as a direct consequence of the genes altered in this region[18,37].

In those patients with these gains, we noted transcriptional changes in several genes associated with aggressive prostate cancer including differential expression of genes appertaining to prognostic assays such as Decipher, Oncotype DX and Prolaris, as well as families such as mucins[50–52]. This may explain the aggressive clinical phenotype observed in these patients. We also observed that when weighted individually, there were differences in enrichment of biological processes between those with *NAALADL2* gains and *TBL1XR1* gains suggesting that each gene results in some unique cellular changes.

Our finding that gains in the 3q26 locus result in concurrent expression of oncogenes located within this region and their downstream targets identifies multiple potential therapeutic avenues warranting further investigations. This study centred around two genes; *NAALADL2* and *TBL1XR1*, both of which are attractive therapeutic targets with *TBL1XR1* previously suggested as a potential cancer target; operating via the TGF-β signalling pathway and potentially regulating AR signalling[53,54]. Additionally, the tumour specificity of *NAALADL2* and basal membranous localisation makes it potentially accessible using antibody-drug conjugates[13]. This approach may be feasible, if like other family members such as PSMA, antibody binding results in subcellular internalisation[12]. Moreover, several of the oncogenes in which gains co-occur, as well as the downstream oncogenes activated from gains in the 3q26 region such as: *ATR*, PI3K family members (*PIK3C2G, PIK3CA, PIK3CB, PIK3R4*), *MUC4, BCL6, SOX9* can be therapeutically targeted or have been suggested as therapeutic targets in cancer[51,55–58]. In the PI3K pathway, *PIK3CB*-specific inhibitors may have utility in patients with mutations, amplifications, and/or fusion of this gene[59]. These findings may have clinical relevence as it has been reported by de Bono et al., that many individuals who had durable (>1 year) responses to *PIK3CB*-specific inhibition harboured activating mutation or amplification in *PIK3CB*[60] and phase II trials of ipatasertib, an Akt inhibitor targeting the PI3K-Akt axis has shown promise in late stage mCRPC[61]. Together, our results suggest that large-scale genomic gains/amplifications occur in the 3q26 region in a high proportion of prostate cancer patients resulting in pro-tumorigenic changes which act to drive cancer aggression. Future work will be required to determine if this genetic alteration can be used to better stratify patients for therapeutics targeting activated downstream pathways.

## Methods

### Patient cohorts

*CNA frequency in prostate cancer subtypes meta-cohort.* A non-overlapping meta-cohort of all available PCa studies (accessed:10/06/2018) were assembled using the cbioportal cancer genomics portal and the alteration frequencies were plotted. A full list of all included studies can be found in supplementary methods. All somatic copy-number estimations had been determined using GISTIC2 or RAE to ensure comparability. At the time of analysis, the total cohort comprised $n = 3804$ patients all included studies listed in Appendix 1. Data downloaded from: https://www.cbioportal.org/.

*ICGC Prostate cancer cohorts.* The ICGC project was launched to coordinate large-scale cancer genome studies in tumours in 50 tumour types using fresh frozen tissue from surgically resected specimens[62]. For CNA analysis, the ICGC prostate cancer Prostate Adenocarcinoma United Kingdom (PRAD-UK) and Prostate Adenocarcinoma Canada (PRAD-CA) both had sufficient available data for GISTIC2 analysis (segment mean files). Some patient ID's were also mapped to the TCGA study therefore, a non-overlapping list of patients resulted in $n = 99$ for ICGC UK and $n = 389$ for ICGC Canada. For the somatic single base alteration comparison, data was pooled from the first two cohorts and the Prostate Adenocarcinoma France (PRAD-FR) cohort ($n = 566$). We also performed similar analysis with the Early Onset Prostate Cancer Germany (EOPC-DE) cohort. Only single base alterations in *NAALADL2* and *TBL1XR1* were included in the comparison to determine reccurenc free survival following radical prostatectomy. Data downloaded on 10/06/2018 from https://dcc.icgc.org/was used for all analyses.

*TCGA prostate cohort.* The TCGA cohort is comprised of $n = 499$ primary prostate carcinomas removed by surgical resection with no prior treatment. 492 patients underwent comprehensive molecular analysis (including somatic copy-number and RNAseq) using fresh frozen tissue. Samples were evaluated by multiple pathologists and cases were excluded if no tumour cells were present or RNA was significantly degraded. The average follow-up time following radical prostatectomy was just under 24 months[9]. The most recent version of the data (TCGA prostate adenocarcinoma, downloaded 10/06/2018) was used for all analyses and downloaded from http://firebrowse.org/?cohort=PRAD&download_dialog=true.

*Statistical analysis and reproducibility.* All statistical analyses were conducted using IBM SPSS statistical analysis software or R version 3.3.1 and visualised using either R version 3.3.1 (packages: Dplyr and ggplot) or IBM SPSS statistics 24 for Windows version 22.0.

*Copy-number estimation.* GISTIC2 (version 6.15.28) analysis was performed in the publicly available GenePattern platform (http://genepattern.broadinstitute.org/) using the settings described by the broad institute[63]. Briefly, GISTIC applies both low- and high-level thresholds to gene copy levels of all patient samples. Those samples which exceed the high-level thresholds $(+/-2)$ are classified as amplifications/deep deletions, and those which exceed the low-level thresholds $(+/-1)$ but not the high-level thresholds are called as gain/shallow deletion. This analysis was ran using seg files containing: sample ID, chromosome name or ID, segment genomic start position, segment end position, number of probes or bins covered by the segment and the seg.mean values.

*Combined estimation of copy-number frequency.* In order to estimate the true frequency of gains in both genes, we employed the use of a random-effects meta-analysis model as described previously[64,65]. Breifly, for the three studies: ICGC UK, ICGC Canada and TCGA the number of men with a gain or amplification in either *NAALADL2* or *TBL1XR1* was extracted as well as the total number of patients in the cohort whom had copy-number assessed. The raw/direct proportions were calculated and we compared the distribution of untransformed, logit and double-arcsine transformed proportions. The distributions of the proportions were assessed for normality using density plots and tested using the Shapiro-Wilk test. Untransformed proportions most resembled a normal distribution (Shapiro-Wilk test of normality: $p = 0.099$) therefore, this transformation was used for the analysis. Due to significant and high inter-study variation demonstrated by a high $I^2$ (84.13%, test for residual heterogeneity: $p = 0.0019$ and 84.2%, test for residual heterogeneity: $p = 0.002$), a random-effects model was fitted. After fitting a random-effects model to all three studies, leave-one-out analyses (LOO) and accompanying diagnostic plots were used to identify influential studies including several measures such as: externally studentized residuals, difference in fits values (DFFITS), Cook's distances, covariance ratios, LOO estimates of the amount of heterogeneity, LOO values of the test statistics for heterogeneity, hat values and weights. In the case of the Canadian ICGC all of these measures identified it as a significant source of heterogeneity therefore it was removed and the model re-fitted. All data analysis and visualisation was performed using the R statistical environment (version 3.6.1, 2019-07-05) using the "metafor" and "meta" packages.

*Co-occurrence of copy number gains/amplifications.* For co-occurrence of CNAs a $2 \times 2$ contingency table was calculated for each gene where imputs were: the number of samples in altered in group 1 (for example, having a gain in *NAALADL2*), the number of samples not altered in group 1 (diploid), compared to the equivalent in group 2 (alteration in gene X) then compared using a Fisher's exact test[66]. All p values were converted to q values to account for false-discovery rate and account for multiple testing (using the qvalue package). A p value of 0.05 or less was considered statistically significant. From the genes identified as co-amplified with *NAALADL2* and *TBL1XR1*, the number of oncogenes was estimated using The Network of Cancer Genes (NCG) tool[30].

*Clinical variable comparrisons.* Chi-squared goodness-of-fit tests were used to compare frequency counts of gains/amplifications between categorical clinical groups such as Grade Group, T stage, tumour location (defined by zone), lymph node positivity and bone scan results. For Grade Group, we also assessed correlation using Spearman's rho, pseudo-coding Grade Groups as a numeric variable and correlating to alteration frequency. All mean comparisons of continuous were carried out using a one-way ANOVA after confirming normality using Q-Q plots and histograms, if necessary variables were Log2 transformed to resemble a more normal distribution. Where results reached the threshold for significance ($p = 0.05$) post-hoc multiple comparisons were assessed using a Tukey test. Age in the TCGA did not have a normal distribution (Shapiro-Wilk test for normality: $p = 0.0087$) so a Kruskal–Wallis test was used to compare age between classes of CNA.

For all Kaplan–Meier plots/survival analysis, univariable analysis was carried out using Log-Rank (Mantel-Cox) and if significant, univariable and/or multivariable Cox regression model was created which included known prognostic factors (see supplementary data). Throughout analysis, no data was excluded or imputed, any information which was not attained was left blank/missing during analysis. Kaplan–Meier estimator curves were constructed using the "survminer" R package.

All described measurements are from distinct samples and no repeated measures were used in this study.

*Identification of androgen regulated DEGs.* To identify DEGs that are potentially androgen regulated we identified those genes harbouring androgen response elements (ARE) from a previously described study[37]. Briefly, palindromic, dihexameric and complete or partial/incomplete AREs were determined by ChIP-Seq analysis. Those genes that also had altered transcriptional expression upon shRNA AR knockdown were considered AR regulated[37]. Here, all genes which themselves were androgen regulated and contained one or more significant motifs were compared to DEGs in patients with either *NAALADL2* or *TBL1XR1* gain/amplifications. Additional genes were identified from a review of a number of studies aimed at identifying AR transcriptionally regulated genes using R1881 treated LNCaPs. This produced list of genes occurring in a minimum of two independent studies[39] and these two resources were compared with the identified DEGs.

*Enrichment analysis.* Gene set enrichment analysis and overrepresentation analysis was performed using webGEstalt (version: 0.4.3, accessed 17/05/2019)[46] analyses were performed using the most recent biological processes GO annotations (GO:BP – releases/2019-03-19). For GSEA, Log2 fold changes computed using DESeq2 were used as the weighting variable. Parameters for analysis were as follows: organism of interest: Homo sapiens, Method of interest: Gene Set Enrichment Analysis or Overrepresentation analysis, Functional database: geneontology (Biological Process noRedundant), minimum number of genes for a single category was set to 5 and maximum to 2000, with up to 1000 permutations for GSEA.

*Data visualisation.* All statistical analyses were conducted using IBM SPSS statistical analysis software and visualised using either R version 3.3.1 (ggplot2, heatmapper). For the enrichment analysis the most significantly altered genes between groups belonging to enriched biological functions were visualised using the "GOPlot" R package.

**Reporting summary**. Further information on research design is available in the Nature Research Reporting Summary linked to this article.

## Data availability
The datasets generated during and/or analysed during the current study are available in the following repositories: Somatic alterations in meta-cohort from cbioportal: https://www.cbioportal.org/, ICGC cohorts: https://dcc.icgc.org/, TCGA PRAD: http://firebrowse.org/?cohort=PRAD&download_dialog=true. Full links are also provided in the materials and methods section. Data used to construct main figures are also provided as supplementary xls files. All relevant data are stored as plain text or xls files and available from the authors upon request, please contact Hayley C. Whitaker (corresponding author) at Hayley.whitaker@ucl.ac.uk.

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

## Acknowledgements

The authors would like to acknowledge the support of University College London, The University of Cambridge, Cancer Research UK. Thanks to the John Black Charitable Foundation, The Urology Foundation and Rosetrees Trust for funding BS and HL. Additionally we are grateful to Prostate Cancer UK and Movember for their support via the Prostate Cancer Centre of Excellence and TLD-PF16-004 grant that supports the work in our laboratory. Finally, we are very grateful to the patients who gave their samples for this research.

## Author contributions

B.S.S. contributed study design, data and statistical analysis, generation of primary hypothesis, wrote manuscript; N.C. conducted additional genomic data analysis and provided bioinformatic expertise; H.L. provided supervision and aided in generation of primary hypothesis; H.P. provided primary data, experimental and data handling guidance, R.F. contributed primary data, genomic and bioinformtic expertise; S.H. provided experimental and biological expertise; J.P. provided primary data and proof-reading of manuscript; C.M.M. contributed supervision and clinical expertise and H.C.W. contributed supervision, aquired funding and aided in generation of primary hypothesis. All authors reviewed and proof-read the manuscript. B.S.S. takes responsibility for veracity of data in this manuscript.

## Competing interests

Dr Niedzica Camacho is an employee of AstraZeneca, Cambridge, UK. The remaining authors declare no competing interests.
