## [Peer Review File · Communications Biology]

Reviewers' comments:

Reviewer #1 (Remarks to the Author):

This study investigates copy number alterations involving NAALADL2 and TBL1XR1 in prostate cancer tumors and their potential association with disease aggressiveness using multiple public databases. Identifying biomarkers that predict aggressive disease is critical for disease treatment, and prior studies have suggested potential importance for this region, making this an interesting and important area of research. However, the study is limited in its ability to draw inferences because of the data used and the lack of additional supporting experimental data. Some of results (e.g., Figure 1) are descriptive, but lack testing to be able to evaluate if the observed differences are real or due to chance. In some cases, the authors state that they tested but do not provide the results; in other cases, it is unknown if tests were conducted. For example, the authors state that NAALADL2 and TBL1XR1 amplifications were highly correlated with grade group (Table 2), but no statistical evidence was presented, making evaluation difficult. The authors look at correlations with gene expression, but don't take the next step to discover what may be driving disease aggressiveness or how disease aggressiveness is being driven. Without additional data, it is hard to tell if CNAs in this region are just a bystander result of aggressive disease or if indeed alterations in this region drive disease aggressiveness. Additional evidence needed to gain a better understanding of the role of this region in disease aggressiveness beyond observed correlations. Finally, there are a few errors/oddities in the manuscript. For example, on Table 1, the percentages for NAALADL2 under TCGA do not add up to 100%. In Figure 2c, there are several large spikes without any shoulder in regions other than 3p26.31 that suggest strong co-occurrence but are not seen in the other datasets and look like possible artifacts.

Reviewer #2 (Remarks to the Author):

Authors of this manuscript propose that CNV in NAALADL2 and TBL1XR1 could serve as prognostic markers for aggressive prostate cancer. While this is an interesting paper, a broad investigation of additional details and questions is recommended:

Major points:

1. Since the authors state that NAALADL2 expression has previously been associated with prostate tumour stage and grade¹³ – authors should be more specific about the novelty of their study (since as they themselves state copy number changes correlate with gene expression)
2. Were other genes considered in this study (as markers of PC aggressiveness)? Why or why not? This needs to be elaborated – the choice of genes might appear somewhat arbitrary....
3. Fig 3 needs a better back-up: since you are showing in Fig 2, that there is a significant association with Gleason grade, the differences in Fig 3 might be simply due to different Gleason scores. To test for this, take the groups from your Kaplan-Meier analysis and plot only GS6, then GS8 etc.... if one of the groups has overwhelming majority of GS6 and another GS8, then the differences in survival are just due to that.... And your findings are due to differences in Gleason, not in survival.... If this is true, conclusions need to be re-stated.
4. Similar (stratified) analysis (as above) is recommended for age groups etc...
- 5/ In addition to the analysis above, it is essential to perform adjusted multivariable Cox, where you

adjust for co-variables, including age, Gleason, treatment etc (otherwise, changes that you find can be simply due to those factors).

6. To re-state: judging from the results, authors presented, it looks to me that there is a strong association with Gleason score – and this might drive the other analysis (unless proven otherwise) – and conclusions might be re-defined.

7. Conclusion “consistent with our hypothesis that transcriptomic changes are being driven by large-scale copy-number changes within the 3q26 locus rather than the functionality of these genes alone” is a bit overstated.... To make this conclusion, perhaps signatures with amplification of NAALADL2 alone, then TBL1XR1 alone, and then amplification of both of them should be compared (or something similar).

8. Markers of these manuscript need to be compared with other known markers of aggressive PC. Do you do better or worse? Or perhaps you rescue different patients?

9. CNV and alterations are used a bit interchangeably.... This needs to be streamlined.

10. What about CNV or alteration frequency in adjacent normal tissues in the prostate (should be available in cBioportal)? This is strongly advised to be reported.

11. Are CNVs of NAALADL2 and TBL1XR1 co-occurring? If yes, then this should be demonstrated.... A heatmap showing their co-occurrence would be useful. Are their co-occurring in the TCGA cohort?

12. “Co-occurrence analysis” with other genes needs to be described better. The number of patients from each cohort, which experienced gene co-occurrence need to be indicated. Also, a heatmap for co-occurrence in each patient across each of the cohorts would help.

13. When defining mRNA signatures for each gene, I would suggest comparing them using GSEA.

14. Methods section needs update: Need to describe the datasets, steps for statistical analysis, why two different tools were used: R or SPSS?

Minor points:

- authors should elaborate a bit more on this statement “NAALADL2 is located on 3q26.31 and closely related to the widely studied PCa marker PSMA12”
- not sure what “alpha level $p = 0.05$ ” is... needs explanation/more details...
- Fisher’s exact test or chi-squared test (or something similar/appropriate) could be used to assess significance of differences in Fig 1.
- something is off in Table 1: 398 corresponds to both 54% and 81%. Might a be a typo. OR perhaps denominators are different? They need to be clearly indicated alongside how these %s were calculated (this is not clear from the table as larger numbers correspond to smaller percents)....
- “Unsupervised hierarchical clustering was applied to the top 50 most significant, differentially expressed genes. Clustering differentiated patients with a NAALADL2 and TBL1XR1 gains from those patients without CNAs.” This is not surprising as this is how you identified these genes to begin with. Thus, I would suggest saying that this is the expected result.
- On page 13, authors mention that identified genes were significantly enriched in cell cycle and mitotic regulation (Fig 4C). Cell cycle is the mother pathway, which encompasses all genes in its child pathway (i.e., mitotic regulation). This needs to be indicated/further investigated/explained.
- Page 12, DESEQ2 should be DESeq2
- The authors should be consistent with Kaplan-Meier Time (either use only months through the paper)

or use only days). Fig 3 and Supplementary Fig 1 need to be inconsistent.
- Version and parameters for GISTIC2 and webGEstalt need to be specified.

Reviewer #3 (Remarks to the Author):

In the manuscript "genetic alterations in the 3q26.31-32 locus confer an aggressive prostate cancer phenotype" by Simpson et al., the authors analyzed copy number alterations of 3q26.31 genes, NAALADL2 and TBL1XR1 in the published studies with near 4000 prostate cancer patients. The authors find that AR their alteration frequency is highly increased in advanced prostate cancer genomes. Further analyses show that the two genes are co-amplified with each other and additional dozens of oncogenes, and their copy number gains correlate with poor prognosis in prostate cancer. Finally, the authors identify a set of differentially expressed genes potentially driven by copy number gains of NAALADL2 and TBL1XR1 in human prostate cancer specimens.

Due to previous reports on TBL1XR1 amplification associated with prostate cancer progression (Heselmeyer-Haddad et al., ref #36; Li et al., ref #42), current study is short of novelty despite providing an unprecedented analysis in big data sets and further confirmed the observation. The same group has evidenced the role of NAALADL2 in prostate tumor growth and progression (Whitaker et al., ref #13), here they may prove experimentally how NAALADL2 works synergistically with TBL1XR1 to promote disease severity. Overall, this is a major concern at this stage, and would need to run experimental assays to discover functional consequence of TBL1XR1, e.g. whether and how its copy gain or overexpression affecting AR cistrome reprogramming (TBL1XR1 and AR, PMIDs: 27129164 and 24243687 that were not cited), and its synergy with NAALADL2 driving prostate cancer.

Some minor comments:

1. The authors highlighted the 3q26.31 locus as an aggressive prostate cancer susceptibility locus. However, whether NAALADL2 is a causal gene warrants further investigation. Relevant to this, in the late part of introduction, rs10936845 was not found to be within a GATA2 motif (Ref #15). The rs10936845 region even showed highest binding preference to HOXB13 but not to GATA2 and FOXA1. The authors should go through the reference in detail and correct the description.
2. On page 10, the authors utilized the Network of Cancer Genes database for prioritizing oncogenes co-occurred with NAALADL2 and TBL1XR1, and should provide concisely the reference for this resource.
3. Though there are only four main figures, figure 3B was not presented in the manuscript.
4. In last results section, it is unclear where the numbers, in particular 447 came from in the sentence "227 of the DEGs were genes from the 447 gene locus we identified (48.8%)". Relevant to this part, did the authors observed overlapped DEGs in their previous cDNA microarray data upon ectopic expression of NAALADL2?
5. In Discussion, "This also supports smaller studies such as those by Heselmeyer-Haddad and colleagues who identified gains in TBL1XR1 in 6 patients with recurrent prostate cancer". In fact, the original study include 7 patients with recurrent prostate cancer.
6. The number of patients in each category was not shown in the supplementary figure 2. To my understanding, the numbers should be the same as these in Table 1. Overall, the Figures are not adequately described in the legend or mentioned in the results text.

Rebuttal general comment

The authors would like to sincerely thank the reviewers for their contributions to this manuscript. The points raised by reviewers have been very beneficial to the manuscript and we are grateful. As a direct consequence of the reviewer comments we have made substantial changes to the manuscript to improve and clarify its content. We feel that these changes satisfy the vast majority of reviewer comments. For ease, we have highlighted the adjustments in yellow in the manuscript.

Reviewer #1:

However, the study is limited in its ability to draw inferences because of the data used and the lack of additional supporting experimental data.

The authors acknowledge our results are of an observational nature, and this is a limitation of our study, as well as the majority of large genomic studies from which we have derived our conclusions. As with these studies, we feel this does not prevent our work from being valuable to the research community and hope that future experimental work will build on the results presented in this paper.

Some of results (e.g., Figure 1) are descriptive, but lack testing to be able to evaluate if the observed differences are real or due to chance. In some cases, the authors state that they tested but do not provide the results; in other cases, it is unknown if tests were conducted. For example, the authors state that NAALADL2 and TBL1XR1 amplifications were highly correlated with grade group (Table 2), but no statistical evidence was presented, making evaluation difficult.

We agree that formal statistical testing would make it easier to evaluate some of our results. We have now added the results of the Chi-squared tests for each of the categorical clinical variables where we have compared the frequency of gains. Using the example you provided, the frequencies of diploid vs gain between grade groups were significantly different from the expected values for both genes ($p=7.844 \times 10^{-08}$ and $p=9.179 \times 10^{-08}$). Moreover, we have now added a much more comprehensive overview of the clinical variables (along with formal statistical testing) to look at the relationship with tumour stage, tumour location, surgical margin status, lymph node positivity and bone scan results, all of which show a statistically significant increase in features associated with aggression. We have added test results elsewhere in the manuscript where they were omitted. We are happy to add any additional tests required by the reviewer.

The authors look at correlations with gene expression, but don't take the next step to discover what may be driving disease aggressiveness or how disease aggressiveness is being driven. Without additional data, it is hard to tell if CNAs in this region are just a bystander result of aggressive disease or if indeed alterations in this region drive disease aggressiveness. Additional evidence needed to gain a better understanding of the role of this region in disease aggressiveness beyond observed correlations.

The authors acknowledge the issue of causation or correlation in studies like ours. We agree that experimental data would add weight to these findings and we hope to build on this study in future, but this was not the focus of our current research. Our group has previously performed in vitro studies of NAALADL2 in isolation and others have looked at TBL1XR1 (PMID: 24240687, 26069883).

Each has been associated individually with tumorigenic processes. Here we aimed to examine these genes (and those surrounding) in a much wider, more physiologically relevant context. Studies such as those by Fraser *et al.*, have conducted exemplary work exploring the genetic alterations of aggressive disease, which in essence, are correlative works (PMID: 28068672). We would suggest that those patients with a gain in this region will possess a number of gene expression changes in well-defined oncogenes whose role is already backed by experimental data (such as MUC1, MUC4 and MUC6 and other prostate cancer associated genes such as SMAD4 and SOX9). This indicates that our conclusions are reasonably justified, albeit not causative.

We also aimed to address this point by conducting an additional investigation; looking for other genetic drivers of aggression such as: PTEN deletion, MYC amplification and other common CNAs which significantly co-occur with NAALADL2/TBL1XR1 gains as they may account for some of our observations. We found no significant co-occurrence with these previously published genetic markers across the three cohorts ($p > 0.05$). We can rule-out that the associations with aggressive clinical features is due to these known genetic drivers of tumour aggression. These findings have now been detailed in the manuscript (located at the bottom of page 9).

Finally, there are a few errors/oddities in the manuscript. For example, on Table 1, the percentages for NAALADL2 under TCGA do not add up to 100%.

Thank you for this comment, there was an error on one of the lines and the figures were also rounded, we have now fixed this and increased the number of significant figures to clarify this result. We have added more detail in this table to facilitate examination of these findings.

In Figure 2c, there are several large spikes without any shoulder in regions other than 3p26.31 that suggest strong co-occurrence but are not seen in the other datasets and look like possible artifacts.

Due to these spikes not validating in the two additional datasets we also believe they are artefacts. This is now stated in the text for clarity.

Reviewer #2 (Remarks to the Author):

Authors of this manuscript propose that CNV in NAALADL2 and TBL1XR1 could serve as prognostic markers for aggressive prostate cancer. While this is an interesting paper, a broad investigation of additional details and questions is recommended

The authors sincerely thank you for this detailed and helpful review of our work. We have worked hard to address these comments. Your comments on the association with Gleason grade were mirrored by other reviewers and to this point, we believe that these genetic changes are intrinsically linked to clinical markers of aggressiveness including Gleason grade. We have therefore conducted a much more thorough investigation into the association with clinical variables and adjusted our conclusions accordingly.

Major points:

1. Since the authors state that NAALADL2 expression has previously been associated with prostate tumour stage and grade – authors should be more specific about the novelty of their study (since as they themselves state copy number changes correlate with gene expression).

This point could be better emphasized, therefore we have expanded this within our discussion. Although our own previous study on NAALADL2 did establish links to aggressive disease this was entirely based on IHC and to a much lesser degree mRNA data. This result was published prior to larger genomic datasets being available. The genetic data provides much greater detail and context about what may be driving this increased expression (such as copy-number alterations). Our more recent investigation has proven that NAALADL2 drives the invasion and migration of PCa cells via the differential expression of genes in vitro. In this study we aimed to focus on the genomic changes to NAALADL2 in cancer and to expand on the phenotypic foundations previously published. Rather than studying this gene in isolation, we have examined the broader context in which this gene becomes overexpressed in aggressive prostate cancer.

Although a number of our observations confirm those made in previous studies given the lack of reproducibility in many biomarker/drug target studies, a major goal of this study was to confirm or refute these previous observations. We have now added this to the discussion to add clarity.

2. Were other genes considered in this study (as markers of PC aggressiveness)? Why or why not? This needs to be elaborated – the choice of genes might appear somewhat arbitrary....

We initially focused on conducting our study centring around NAALADL2, based on previous (and ongoing) research conducted by our group which had already demonstrated the roles of this gene in a tumour development (PMID: 24240687). Additionally, the only known germline alteration predicting a higher Gleason grade occurs in this region, which was curious (PMID: 25939597). Our investigation showed such high co-occurrence with neighbouring oncogene TBL1XR1 that we felt it inappropriate to consider them separately and decided to centre our research on this region. We have now elaborated on this in the manuscript to justify our decision. Given the number of genes with gains in this region we cannot attribute the observations to any individual genes and we have altered the conclusions to reflect this.

3. Fig 3 needs a better back-up: since you are showing in Fig 2, that there is a significant association with Gleason grade, the differences in Fig 3 might be simply due to different Gleason scores. To test for this, take the groups from your Kaplan-Meier analysis and plot only GS6, then GS8 etc.... if one of the groups has overwhelming majority of GS6 and another GS8, then the differences in survival are just due to that.... And your findings are due to differences in Gleason, not in survival.... If this is true, conclusions need to be re-stated.

4. Similar (stratified) analysis (as above) is recommended for age groups etc...

5/ In addition to the analysis above, it is essential to perform adjusted multivariable Cox, where you adjust for co-variates, including age, Gleason, treatment etc (otherwise, changes that you find can be simply due to those factors).

6. To re-state: judging from the results, authors presented, it looks to me that there is a strong association with Gleason score – and this might drive the other analysis (unless proven otherwise) – and conclusions might be re-defined.

The authors agree with points 3-6 and we have chosen to address these collectively as they are thematically similar. The genetic alterations we see associate with a number of macroscopic and observable clinical traits (including Gleason grade) that likely account for the differences in disease-free survival. We had included these Kaplan Meier curves to demonstrate that you can segregate patients with differing survival profiles based solely on their CNA status. We have taken these reviewers points on board and have now added a much more comprehensive overview of the association with clinical variables (now: "NAALADL2 AND TBL1XR1 gains are associated with adverse clinical characteristics") along with formal statistical testing demonstrating that patients with gains in these genes have statistically significant differences in Gleason grade, tumour stage, extraprostatic extension, lymph node positivity, response to initial therapy and bone scan results. Multivariate Cox regression confirmed these observations (now added in text). Finally, we also compared age between groups but found no significant differences (Kruskall-Wallis $p= 0.1191$ and 0.227 for NAALADL2 and TBL1XR1). All of these results have been combined and a new section has been added to the manuscript.

7. Conclusion "consistent with our hypothesis that transcriptomic changes are being driven by large-scale copy-number changes within the 3q26 locus rather than the functionality of these genes alone" is a bit overstated.... To make this conclusion, perhaps signatures with amplification of NAALADL2 alone, then TBL1XR1 alone, and then amplification of both of them should be compared (or something similar).

13. When defining mRNA signatures for each gene, I would suggest comparing them using GSEA.

These two points are related therefore we have chosen to address these points together. The authors are grateful for this suggestion. We have now performed GSEA of both genes separately as well as the ORA of the combined DEGs. Curiously, despite a 77.9% overlap in DEGs, this analysis revealed some distinct differences in the enriched biological processes between NAALADL2 and TBL1XR1, potentially indicating different functional consequences for gains in one versus another. As the majority of patients had gains in both together, we may gather any effect is a combination of these enriched functions. We have now added these results into the manuscript as well as the files for full transparency and adjusted our conclusions.

8. Markers of these manuscript need to be compared with other known markers of aggressive PC.

We have looked at co-occurrence of NAALADL2/TBL1XR1 gains/amplifications with other known markers of aggressive disease in other loci such as MYC gain, FGFR1 gain, PTEN loss, RB1 loss or NKX3-1 loss. We found no significant overlap across the three datasets, indicating that while some patients possess multiple CNAs, generally speaking CNAs in these key genes are in different patients. This has been added text and a genome-wide co-occurrence table with q values is provided in supplementary).

Do you do better or worse? Or perhaps you rescue different patients?

For reference to our results we have added Kaplan–Meier estimator curves for patients with and without PTEN loss and MYC gain (selected as they are perhaps the most well described within the literature) showing the disease-free survival of patients with and without these alterations (supplementary figure 3). When separated purely on the basis of copy-number status, patients with

gains in NAALADL2 and TBL1XR1 have worse outcomes than those with PTEN loss or MYC gain. We also looked at the disease-free survival of patients with and without these CNAs where the patient groups were mutually exclusive (for example: if possessing a NAALADL2/TBL1XR1 gain, could not have PTEN loss) and found that NAALADL2/TBL1XR1 predicted worse or similar outcome to PTEN/MYC alterations. Patients with CNAs in both NAALADL2/TBL1XR1 and PTEN/MYC had the worst survival. We have added this to the text below Figure 3 and is shown in Supplementary Figure 3.

9. CNV and alterations are used a bit interchangeably.... This needs to be streamlined.

The term 'alteration' was used to refer to either gain or amplification however, we agree this can be ambiguous and may refer to copy-number loss. We have now changed this to gain/amplification unless we are intentionally being ambiguous.

10. What about CNV or alteration frequency in adjacent normal tissues in the prostate (should be available in cBioportal)? This is strongly advised to be reported.

All of the CNAs reported are somatic and called relative to normal tissue, this has now been clarified in-text.

11. Are CNVs of NAALADL2 and TBL1XR1 co-occurring? If yes, then this should be demonstrated.... A heatmap showing their co-occurrence would be useful. Are their co-co-occurring in the TCGA cohort?

12. "Co-occurrence analysis" with other genes needs to be described better. The number of patients from each cohort, which experienced gene co-occurrence need to be indicated. Also, a heatmap for co-occurrence in each patient across each of the cohorts would help

To address both these points, we have now added the co-occurrence matrix for each study showing the correlation between different classes of NAALADL2 or TBL1XR1 CNA as Supplementary Figure 1 and described the co-occurrence in more detail in the methods section.

14. Methods section needs update: Need to describe the datasets, steps for statistical analysis, why two different tools were used: R or SPSS?

We have added more detail to the methods and to the cohort information. The use of both SPSS and R was simply as a result of different researchers having preferred methods for data visualisation and different levels of familiarity with software. We have now added much more detail to the methods section as requested.

Minor points:

- authors should elaborate a bit more on this statement "NAALADL2 is located on 3q26.31 and closely related to the widely studied PCa marker PSMA12"

This has now been added to the text.

- not sure what "alpha level $p = 0.05$ " is... needs explanation/more details...

This was to refer to the significance level set prior to testing. This is inferred from the start of the sentence, so has been removed from the text.

- Fisher's exact test or chi-squared test (or something similar/appropriate) could be used to assess significance of differences in Fig 1.

We have now performed this test and reported the results.

- something is off in Table 1: 398 corresponds to both 54% and 81%. Might a be a typo. OR perhaps denominators are different? They need to be clearly indicated alongside how these %s were calculated (this is not clear from the table as larger numbers correspond to smaller percents).....

Apologies, this was an error in the table (did not effect the values reported in text), we have now rectified this and added the totals to make the table more clear.

- "Unsupervised hierarchal clustering was applied to the top 50 most significant, differentially expressed genes. Clustering differentiated patients with a NAALADL2 and TBL1XR1 gains from those patients without CNAs." This is not surprising as this is how you identified these genes to begin with. Thus, I would suggest saying that this is the expected result.

The authors have added this suggestion into the main text.

- On page 13, authors mention that identified genes were significantly enriched in cell cycle and mitotic regulation (Fig 4C). Cell cycle is the mother pathway, which encompasses all genes in its child pathway (i.e., mitotic regulation). This needs to be indicated/further investigated/explained.

This has now been added to the text.

- Page 12, DESEQ2 should be DESeq2

This has now been corrected in text.

- The authors should be consistent with Kaplan-Meier Time (either use only months through the paper or use only days). Fig 3 and Supplementary Fig 1 need to be inconsistent.

This has now been added to the figure.

- Version and parameters for GISTIC2 and webGEstalt need to be specified.

We have added the versions and more detail on this in our methodology.

Reviewer #3 (Remarks to the Author):

Due to previous reports on TBL1XR1 amplification associated with prostate cancer progression (Heselmeyer-Haddad et al., ref #36; Li et al., ref #42), current study is short of novelty despite providing an unprecedented analysis in big data sets and further confirmed the observation. The same group has evidenced the role of NAALADL2 in prostate tumour growth and progression

(Whitaker et al., ref #13), here they may prove experimentally how NAALADL2 works synergistically with TBL1XR1 to promote disease severity. Overall, this is a major concern at this stage, and would need to run experimental assays to discover functional consequence of TBL1XR1, e.g. whether and how its copy gain or overexpression affecting AR cistrome reprogramming (TBL1XR1 and AR, PMIDs: 27129164 and 24243687 that were not cited), and its synergy with NAALADL2 driving prostate cancer.

The Heselmeyer-Haddad group performed their analysis in a cohort of seven patients with recurrent tumours whereby they found 29% of patients had a gain in this region. This translates to just two patients, which did not allow for formal statistical testing and we cannot be confident that this result did not occur by chance. The Heselmeyer-Haddad cohort was preselected to represent 'recurrent disease', defined by relatively loose criteria of "two consecutive PSA measurements within 1 year of ≥ 0.2 ng/mL and/or evidence of metastatic disease.". Therefore, while interesting, the Heselmeyer-Haddad study lacks power in its observations and generalisability. The same study looked at these genes in isolation and did no further downstream analysis to determine which transcriptomic pathways or biological functions were activated in these patients, and could not complete a robust comparison of the clinical associations with this alteration due to small patient numbers.

The Li *et al.*, review article discusses TBL1XR1's association with aggressive clinical features in some tumour types but also highlights the need for our study. In some cancers such as acute lymphoblastic leukemia, deletions of TBL1XR1 are significantly more common in patients who relapse. Indeed in prostate cancer we currently have conflicting evidence on whether TBL1XR1 acts as a tumour suppressor or activator as studies such as the one by Daniels *et al.*, showed lower expression in malignant tissue and seemed to decrease prostate cancer growth (PMID: 24243687). We acknowledge the reviewers point regarding AR interactions/cistrome reprogramming however, this was not the focus of our study and we could not provide an adequate investigation into this process using these cohorts. The authors agree this hypothesis is interesting, so we have added a breakdown of the DEGs which are known to be androgen regulated and those which contain AR binding motifs, to aid future investigations.

Our own previous study on NAALADL2 did establish links to aggressive disease, but this was entirely based on IHC and to a much lesser degree mRNA data. This result was published prior to larger genomic datasets being available. The genetic data provides much greater detail and context about what may be driving this increased expression (such as copy-number alterations). Our more recent investigation has proven that NAALADL2 drives the invasion and migration of PCa cells via the differential expression of genes in vitro. In this study we aimed to focus on the genomic changes to NAALADL2 in cancer and to expand on the phenotypic foundations previously published. Rather than studying this gene in isolation, we have examined the broader context in which this gene becomes overexpressed in aggressive prostate cancer. For these reasons we feel our study is justified and represents significant progress. We have also added several entirely novel findings to the current evidence including: association with neuroendocrine and castrate resistant prostate cancer, clinical association with Gleason grade, co-occurrence with neighbouring oncogenes and the activation of transcriptomic profiles indicative of increased cell proliferation. Moreover, in the process of refining this manuscript during peer-review we have added additional data which adds further novelty.

Some minor comments:

1. The authors highlighted the 3q26.31 locus as an aggressive prostate cancer susceptibility locus. However, whether NAALADL2 is a causal gene warrants further investigation.

The authors feel that our previous investigation on NAALADL2 did as much as possible to ascertain the isolated role of this protein in cancer cells. In this study, our aim was to explore possible causes of mRNA and protein expression but more importantly, to look at the gene in a larger physiological context. As we found that changes in this gene do not exist in isolation, therefore direct causal inferences were not made in the present study.

Relevant to this, in the late part of introduction, rs10936845 was not found to be within a GATA2 motif (Ref #15).

Please see Supplementary figure S1 entitled: "Association between rs10936845 genotype, GATA2 motif, NAALADL2 expression and PCa risk." A canonical GATA2 motif is shown whereby the A allele at rs10936845 (incorrectly labelled as rs6057110 within the legend) disrupts a canonical GATA2 motif.

The rs10936845 region even showed highest binding preference to HOXB13 but not to GATA2 and FOXA1. The authors should go through the reference in detail and correct the description.

We agree that the rs10936845 region showed increased binding preference to HOXB13, with lower binding affinity to FOXA1 and GATA2. Jin *et al.*, had interpreted their results as evidence of co-occupancy, we had highlighted GATA2 due to the significant association with NAALADL2 expression and the allele genotypes. We have now expanded the description.

2. On page 10, the authors utilized the Network of Cancer Genes database for prioritizing oncogenes co-occurred with NAALADL2 and TBL1XR1, and should provide concisely the reference for this resource.

A reference for this resource was provided in the materials and methods section, this has now been added to the results section of the manuscript.

3. Though there are only four main figures, figure 3B was not presented in the manuscript.

This has now been rectified.

4. In last results section, it is unclear where the numbers, in particular 447 came from in the sentence "227 of the DEGs were genes from the 447 gene locus we identified (48.8%)".

This number should have been 465, representing the number of statistically co-occurring genes which were consistent across all three cohorts. We have now rectified this and clarified how this number was derived.

Relevant to this part, did the authors observed overlapped DEGs in their previous cDNA microarray data upon ectopic expression of NAALADL2?

In our previous study identified 9 genes which were reciprocally regulated by overexpression or knockdown of NAALADL2. Of these 9 we found that 3 (cancer antigen XAGE1B, adhesion/motility regulator SPON2 and AR regulator HN1) were significantly differentially expressed ($p < 0.022$) in

patients with a NAALADL2 gain and in the same direction as the overexpression model. We have now reported this in the text.

5. In Discussion, “This also supports smaller studies such as those by Heselmeyer-Haddad and colleagues who identified gains in TBL1XR1 in 6 patients with recurrent prostate cancer”. In fact, the original study include 7 patients with recurrent prostate cancer.

This has now been rectified.

6. The number of patients in each category was not shown in the supplementary figure 2. To my understanding, the numbers should be the same as these in Table 1. Overall, the Figures are not adequately described in the legend or mentioned in the results text.

This is correct, the numbers are exactly as in Table 1. We have expanded the legend and results to provide more detail.

Reviewers' comments:

Reviewer #1 (Remarks to the Author):

The authors have addressed my concerns. Some clarification is needed regarding the FDR values presented. It is not clear if these are FDR-adjusted p-values or what the false discovery rate is based on. Since the datasets are not large and estimates not precise, the authors should limit the number of significant digits presented for their results.

Reviewer #2 (Remarks to the Author):

The authors did a good job addressing my comments and have clarified several important points in the manuscript. However, in light of new data and analyses, I have additional comments to streamline the paper:

Overall comment:

Based on the additional provided data/results, I would recommend centering the paper around Gleason (and other relevant variables), not disease-free survival. As the authors now show that adjusting for Gleason and T stage, genes of interest lost their prognostic ability, Gleason should be the central point of this manuscript. Given the new data, KM analysis is misleading and is recommended to be removed. Instead, Gleason should be used as a main outcome (not disease-free survival).

Specific major points:

1. The result "As CNAs in NAALADL2 and TBL1XR1 were associated with clinical characteristics such as Gleason grade group and T stage, we used multivariate Cox regression models to confirm that any changes in survival were driven by these associations and found that copy number gains in NAALADL2 and TBL1XR1 were no longer significant once corrected for Gleason grade and T stage ($p= 0.71184$, Supplementary file 3). These results suggest that the differences in disease-free survival seen when stratified by gain/amplification status are driven by strong association with these clinical variables." means that Gleason is driving the survival analysis and since the genes of interest correlate with Gleason, they do not drive the survival analysis themselves (do not have independent prognostic value). In light of these results, I would suggest removing KM survival analysis (since what we see is basically separation between Gleasons).

Alternative analysis I can suggest (to possibly rescue the KM analysis) is to break the cohorts into Gleason grade groups (Gleason 1+2, Gleason 3, etc – some Gleasons might need to be grouped). And inside each group, you can investigate if patients with gains/amplifications perform are separated in KM analysis.

2. Could Pten and MYC stratify Gleason groups? Are they doing better than the genes of interest? Would be great to see a heatmap with genomic alterations for genes of interest, Myc, and Pten across Gleason scores.

3. Statement "Leave-one out analysis revealed that the ICGC Canada study may represent an outlier." Is a bit confusing.... "outlier" needs to be defined/explained in this specific context. Do you mean that ICGC Canada is very different from TCGA and ICGC UK? Maybe you simply see a batch effect? In that case, you SHOULD NOT combine these datasets and they should be analyzed separately (should not be combined with others).

4. Univariate meaning one outcome is measured. Univariable means that you use one input variable. Same for multivariable/multivariate (multivariate= multiple outcomes).

5. "Unsupervised hierarchical clustering of the top 50 most significant, shared DEGs. Clustered together genes that differentiated patients with NAALADL2 and TBL1XR1 gains from those patients without CNAs." – this result is expected as this is how you identified the differentially expression genes to begin with. Thus, this is a circular argument.

6. FDR=0 should not be reported. There is a minimum p-value estimated by the GSEA (which depends on how many times you do permutations). That p-value should be used as input to FDR correction. Then FDR<x (where x is the smallest possible after the correction) should be reported.

7. Statistical test need to be reported throughout the paper. Whenever p-values are calculated and reported, the test needs to be indicated in front of it.

Specific minor comments:

1. "These genetic gains associate with reduced disease-free survival after radical prostatectomy." should be down-played/removed (given your findings on the strong association with Gleason)

2. In a sentence "Positional changes are also known to alter transcriptional regulation" – "positional changes" need to be defined and it should be explained why and how positional changes could affect transcriptional regulation.

3. "Increased NAALADL2 and TBL1XR1 expression have previously been linked to poor prognosis in cancers leading us to examine the frequency of somatic copy-number gains in these genes across various prostate cancer subtypes." – needs citations

4. "Chi-squared $p= 1.19 \times 10^{-6}$ and $p= 2.39 \times 10^{-6}$ " –Are these values for Fig 1A and for Fig 1B? Was chi-square goodness of fit test used or chi-square test for independence? (I assume goodness of fit). This needs to be clarified and indicated throughout the paper.

5. Fig 1B is never referenced in the text (please check for all Figures).

6. This is a bad name for the table "Known oncogenes from DESEQ2" (also normalization technique should be corrected in the table as well) – it does not reflect the message for this analysis. All table names and tables should be streamlined.

7. In the statement "In primary prostate cancer, CNAs in this region associated with Gleason grade, tumour stage, number of positive lymph nodes, bone scan results and a reduction in disease-free survival" – reduction of disease-free survival should be removed as it is significantly driven by Gleason.

Reviewer #3 (Remarks to the Author):

The authors have extensively revised the manuscript, involving additional data analysis and further supporting results provided, as well as convincing explanation and back-up of the novelty of the study. This reviewer agrees that the genomic study will leave interests to future investigation. I thus feel

happy for the manuscript that can be accepted by Communications Biology if the feedback and comments from the other reviewer are unanimous.

Reviewer #1 (Remarks to the Author):

The authors have addressed my concerns. Some clarification is needed regarding the FDR values presented. It is not clear if these are FDR-adjusted p-values or what the false discovery rate is based on. Since the datasets are not large and estimates not precise, the authors should limit the number of significant digits presented for their results.

Thank you for your useful suggestions. We can confirm that these are FDR-adjusted p-values and this has been added to text. We have reduced the significant figure to 2 based on Nature author guidelines.

Reviewer #2 (Remarks to the Author):

The authors did a good job addressing my comments and have clarified several important points in the manuscript. However, in light of new data and analyses, I have additional comments to streamline the paper:

Thank you for another round of useful suggestions, again adding value to our manuscript. The points you raise are interesting and we have done our best to integrate these suggestions in the manuscript.

Overall comment:

Based on the additional provided data/results, I would recommend centering the paper around Gleason (and other relevant variables), not disease-free survival. As the authors now show that adjusting for Gleason and T stage, genes of interest lost their prognostic ability, Gleason should be the central point of this manuscript. Given the new data, KM analysis is misleading and is recommended to be removed. Instead, Gleason should be used as a main outcome (not disease-free survival).

Thank you, we have done our best to incorporate these suggestions. We hope the article is now suitable for publication.

Specific major points:

1. The result "As CNAs in NAALADL2 and TBL1XR1 were associated with clinical characteristics such as Gleason grade group and T stage, we used multivariate Cox regression models to confirm that any changes in survival were driven by these associations and found that copy number gains in NAALADL2 and TBL1XR1 were no longer significant once corrected for Gleason grade and T stage ($p= 0.71184$, Supplementary file 3). These results suggest that the differences in disease-free survival seen when stratified by gain/amplification status are driven by strong association with these clinical variables." means that Gleason is driving the survival analysis and since the genes of interest correlate with Gleason, they do not drive the survival analysis themselves (do not have independent prognostic value). In light of these results, I would suggest removing KM survival analysis (since what we see is basically separation between Gleasons).

Thank you, we agree that these clinical associations certainly explain the differences in survival and have now moved these into supplementary. We have not however, removed the survival analysis altogether as it shows that by stratifying patients purely by NAALADL2/TBL1XR1 status you can see differences in survival (due to their strong association with a number of important clinical features) and this is likely a result of all these clinical variables. For example, T stage was not significant in the Cox regression however it is unlikely that it doesn't contribute at all. PTEN and MYC have a similar association with adverse prognostic features (reported in other papers) and in the TCGA cohort we found that the Spearman correlation with Gleason grade group was still high (rho was 0.6 and 0.7) however, as shown in supplementary figure 4 they do not bifurcate the survival profiles as strongly.

We have now clarified this in-text in the results section and in the discussion to avoid misleading the reader.

Additionally, the other viewers were satisfied with the inclusion of this data. We hope this decision is acceptable to the reviewer and we have been careful to not imply these genes are driving these changes in survival in-text.

Alternative analysis I can suggest (to possibly rescue the KM analysis) is to break the cohorts into Gleason grade groups (Gleason 1+2, Gleason 3, etc – some Gleasons might need to be grouped). And inside each group, you can investigate if patients with gains/amplifications perform are separated in KM analysis.

The authors thank you for this suggestion however, after separating by Gleason grade groups we could not see strong enough differences to warrant inclusion in the paper.

2. Could Pten and MYC stratify Gleason groups? Are they doing better than the genes of interest? Would be great to see a heatmap with genomic alterations for genes of interest, Myc, and Pten across Gleason scores.

The two genes of interest have a strong correlation (Spearman) to Gleason grade group rho = 0.9, p=0.0374 and 0.9, p=0.0374, compared to PTEN 0.6, p=0.2848, and MYC 0.7 p=0.1881.

We attempted a heatmap to show this however, as the copy-number values are either binary (gain/no gain) or as categories of alteration; deep deletion, shallow deletion, diploid etc and Gleason grade is also categorical the heatmap didn't look very clear. We have attached a heatmap clustered with Euclidian distance (left) and one where the row is ordered by Gleason Grade group (right) as examples. In the heatmap where row order is forced (right) it somewhat demonstrates that increased frequency is seen in higher Gleason in all four genes, however as PTEN and MYC are generally more frequent (even though the frequency correlates more poorly to Gleason) they somewhat dominate the heatmap which could be misleading to the reader.

Would the reviewer be satisfied with simply reporting of the correlation coefficients in-text? Alternatively, if there is a more suitable way to create this heatmap or the reviewer would like the heatmaps included as supplementary we are happy to do so.

3. Statement “Leave-one out analysis revealed that the ICGC Canada study may represent an outlier.” Is a bit confusing....”outlier” needs to be defined/explained in this specific context. Do you mean that ICGC Canada is very different from TCGA and ICGC UK? Maybe you simply see a batch effect? In that case, you SHOULD NOT combine these datasets and they should be analyzed separately (should not be combined with others).

After fitting a random-effects model to all three studies, leave-one-out analyses (LOO) and accompanying diagnostic plots were used to identify influential studies including several measures such as: externally studentized residuals, difference in fits values (DFFITS), Cook’s distances, covariance ratios, LOO estimates of the amount of heterogeneity, LOO values of the test statistics for heterogeneity, hat values and weights. In the case of the Canadian ICGC all of these measures identified it as an outlier.

We had initially chosen not to exclude it as with only three studies, we could not be sure that this was a true outlier or simply reflected the fact that the ICGC UK and TCGA values were very similar and we have a small number of included studies.

We have now re-fitted the model without the ICGC Canada study and reported the new values in the text. We have increased the level of detail in the methods section to aid clarity.

4. Univariate meaning one outcome is measured. Univariable means that you use one input variable. Same for multivariable/multivariate (multivariate= multiple outcomes).

Thank you, we have now reworded this in-text.

5. “Unsupervised hierarchal clustering of the top 50 most significant, shared DEGs. Clustered

together genes that differentiated patients with NAALADL2 and TBL1XR1 gains from those patients without CNAs.” – this result is expected as this is how you identified the differentially expression genes to begin with. Thus, this is a circular argument.

Thank you, we have now removed this line from the manuscript.

6. FDR=0 should not be reported. There is a minimum p-value estimated by the GSEA (which depends on how many times you do permutations). That p-value should be used as input to FDR correction. Then $FDR < x$ (where x is the smallest possible after the correction) should be reported.

Thank you, we have now removed this line from the manuscript and used “FDR < 0.0001”.

7. Statistical test need to be reported throughout the paper. Whenever p-values are calculated and reported, the test needs to be indicated in front of it.

We appreciate this was unclear. Where appropriate we have now included the name of the test next to all p-values.

Specific minor comments:

1. “These genetic gains associate with reduced disease-free survival after radical prostatectomy.” should be down-played/removed (given your findings on the strong association with Gleason)

We have now changed this to emphasize the association with clinical variables. In-text and in the discussion we have now down-played the significance of survival differences.

2. In a sentence “Positional changes are also known to alter transcriptional regulation” – “positional changes” need to be defined and it should be explained why and how positional changes could affect transcriptional regulation.

We have now clarified this and elaborated this in-text.

3. “Increased NAALADL2 and TBL1XR1 expression have previously been linked to poor prognosis in cancers leading us to examine the frequency of somatic copy-number gains in these genes across various prostate cancer subtypes.” – needs citations

Thank you, we have now added in these citations.

4. “Chi-squared $p = 1.19 \times 10^{-6}$ and $p = 2.39 \times 10^{-6}$ ” –Are these values for Fig 1A and for Fig 1B? Was chi-square goodness of fit test used or chi-square test for independence? (I assume goodness of fit). This needs to be clarified and indicated throughout the paper.

You are correct these are for Fig1A&B and this was goodness-of-fit. We have now corrected this in-text and elaborated in each instance where this test is used.

5. Fig 1B is never referenced in the text (please check for all Figures).

Thank you, we have now added this into the text.

6. This is a bad name for the table “Known oncogenes from DESEQ2” (also normalization technique should be corrected in the table as well) – it does not reflect the message for this analysis. All table names and tables should be streamlined.

Thank you, we have renamed this list “Differentially expressed oncogenes”. Hopefully, you feel this better reflects the message. We have also tried to clarify the title and contents of the supplementary files.

The supplementary file names are now as follows:

“Supplementary_file_1_Gain_Co-occurrence_in_3_cohorts”

“Supplementary_file_2_Co-amplified_oncogenes”

“Supplementary_file_3_Multivariable_Cox_regressions_TCGA”

“Supplementary_file_4_DEG_analysis_DESeq2”

“Supplementary_file_5_Oncogenes_present_in_DEG_list_overlap”

“Supplementary_file_6_GSEA_for_NAALADL2_TBL1XR1_DEGs”

“Supplementary_file_7_ORA_Analysis_for_shared_DEGs”

7. In the statement “In primary prostate cancer, CNAs in this region associated with Gleason grade, tumour stage, number of positive lymph nodes, bone scan results and a reduction in disease-free survival” – reduction of disease-free survival should be removed as it is significantly driven by Gleason.

We have now removed this from the text.

Reviewer #3 (Remarks to the Author):

The authors have extensively revised the manuscript, involving additional data analysis and further supporting results provided, as well as convincing explanation and back-up of the novelty of the study. This reviewer agrees that the genomic study will leave interests to future investigation. I thus feel happy for the manuscript that can be accepted by Communications Biology if the feedback and comments from the other reviewer are unanimous.

The authors would like to thank you for your time, input and ideas, we are pleased you are satisfied with our changes.

REVIEWERS' COMMENTS:

Reviewer #2 (Remarks to the Author):

The manuscript has been largely improved and thought through. Authors tried to address my comments to the best of their abilities.